# Gender Differences in Kinematic Parameters of Topspin Forehand and Backhand in Table Tennis

**DOI:** 10.3390/ijerph17165742

**Published:** 2020-08-08

**Authors:** Ziemowit Bańkosz, Sławomir Winiarski, Ivan Malagoli Lanzoni

**Affiliations:** 1Department of Biomechanics, Faculty of Physical Education and Sports, University School of Physical, Education in Wrocław, 51-612 Wrocław, Poland; slawomir.winiarski@awf.wroc.pl; 2Department for Life Quality Studies, University of Bologna, 47921 Rimini, Italy; ivan.malagoli@unibo.it

**Keywords:** gender differences, kinematics, table tennis, sports technique

## Abstract

*Background:* The identification of gender differences in kinematics and coordination of movement in different body segments in sports may improve the training process by emphasizing the necessity of its differentiation, and consequently individualization, developing, and improving the technique in women and men. Indicating differences can also help in determining the risk of injury in order to prevent from them by diversifying training programs. However, there is no information regarding this problem in the existing literature pertaining to table tennis. Therefore, the aim of the study was to evaluate the differences in the values of selected angular and kinematic parameters during topspin forehand and topspin backhand shots between male and female table tennis players. *Methods:* Six male and six female advanced table tennis players performed topspin forehand and topspin backhand shots, both receiving a backspin ball. The angular parameters in four events (ready position, backswing, maximum acceleration, and forward) at chosen joints as well as the maximal acceleration of the playing hand were measured, using the myoMotion system, and were compared between male and female players. *Results:* Significant differences (*p* ≤ 0.05) were found in the magnitude of angular parameters and maximum hand acceleration between men and women. The movement pattern of topspin strokes performed by men takes into account, more than that in the case of women, movements that use large muscle groups and large joints (hip joints, trunk joints, shoulder joints in extension, and flexion). The difference in the values of maximal acceleration reached almost 50 m/s^2^ in topspin forehand (*p* < 0.01) and 20 m/s^2^ in backhand (*p* < 0.01). *Conclusions:* Differentiation of movement patterns can be a manifestation of movement optimization due to anthropological differences and limitations. The differences in the values of maximal acceleration suggest that women could use both sides to perform a topspin attack against the backspin ball, while men should seek opportunities to make a stronger shot with a forehand topspin.

## 1. Introduction

Various studies have highlighted the benefits of playing table tennis as a form of recreation and leisure, such as improving hand-eye coordination [1], improving balance, coordination, brain stimulation and development of cognitive functions [2], development of body build, and improving fat distribution [3]. Furthermore, as a sport practiced by professional players, table tennis is extremely demanding. The skill level in this sport is determined by a great number of factors, which are combined in terms of physical preparation (fitness and coordination aptitudes), technical preparation (e.g., perfection, variability, and variety of playing techniques), tactical preparation (e.g., planning and “reading” the game and adjustment), and mental preparation (e.g., positive attitude, attention, level of emotions, etc.) [4,5]. Providing opportunities for sustainable development of all skills in the above-mentioned areas seems to be a very important aspect of the training process. The basic principles of training involves individualization, the aim of which is to adjust the load and training programs to the various individualized needs of the athlete [6]. Their diversity may result not only from differences in anatomical body build, level of development of motor skills but also from age, gender, level of technology, or psychological determinants. The diversity described may be manifested in the variety of techniques, characterized by the different movements in joints and kinematic or angular parameters. Differentiation of kinematic parameters in table tennis has been explored in previous studies. Bańkosz and Winiarski pointed out high inter- and intra-individual variability of kinematic parameters of topspin forehand in table tennis, suggesting the existence of movement functionality and functional variability [7]. The authors also concluded that according to the phenomenon of equifinality, even though the players used different methods of performing the movement, they obtained similar values of acceleration of playing hand. The implication of above findings includes the necessity of individualized training programs. An interesting issue that has not been addressed to date in the literature pertaining to table tennis is the diversity of techniques in the sport in relation to the gender of the players. Gender differences in table tennis were shown so far in morphological structure. It was found that male table tennis players have higher fat-free body mass and fat mass percent indices than female players [8]. Assessment of typical gender differences in the details of the technique, such as kinematic or physiological parameters, may allow for setting the coaching objectives, optimizing and preparing appropriate training plans, tailored to the gender of the athlete, and developing the technique or preventing the risk of injury [9,10,11]. Gender differences in kinematics have been shown in many sports. Young runners have been found to have a significant gender effect on running mechanics [12,13,14]. McLean et al. found differences in the kinematics of the knees, hips, and ankles between men and women who play basketball [15]. Due to these differences, the authors also stressed upon the higher risk of anterior cruciate ligament injury (ACL) while playing in women. Another study found differences in the trunk and pelvic kinematics between female and male young rowers [16]. The authors suggested potentially different biomechanical loading mechanisms in rowing in women and men. Similarly, gender differences in the kinematics of hips and knee joints during a quick start were found in hockey players [17]. It was also found that there are differences in kinematics between women and men in martial arts [18]. It seems interesting to answer the question of whether there are differences in the technique of performing table tennis shots between men and women. The findings presented in the literature show significant differences between males and females in body composition, proportion of body segments, etc. [8,19], which could cause other functional differences. Their identification may improve the training process by emphasizing the necessity of differentiation, and consequently individualization, in developing and improving the technique in women and men. Indicating differences in kinematics and coordination of movement in different body segments (the angular position at joints during the movement, especially in main events) can also help in determining differences in relation to the risk of injury in order to prevent from them by diversifying training programs. Topspin forehand and topspin backhand in modern table tennis are the most commonly used, aggressive, offensive shots, opening the offensive rally or bringing directly the point [20]. They are the fastest and most aggressive, especially when receiving backspin ball [21]. Therefore, the aim of the study was to evaluate the differences in the values of selected angular and kinematic parameters during topspin forehand and topspin backhand shots between men and women in table tennis. Due to differences in body composition and proportion of body muscles and mass, it was assumed that there are differences between men and women in the values of angular parameters and the way in which movement in selected joints is performed, and therefore in the values of hand acceleration. These differences concern the use of the trunk and shoulder when generating force to perform topspin forehand and backhand, which seems to be larger in males than in females.

## 2. Material and Methods

The research involved six male and six female advanced (national team level) table tennis players of 22.9 ± 2.8 and 21.1 ± 1.5 y age, body height 178 ± 2.5 and 165 ± 2.5 cm, and body mass 77 ± 7.5 and 59 ± 4.5 kg for men and women, respectively. All participants were informed about the research aim and provided informed consent to participate in the experiment. The Research Bioethics Commission of The Senate Bioethics Research Committee of University School of Physical Education in Wrocław approved the experiment (34/2019).

The participants performed two tasks: topspin forehand (TF) and topspin backhand (TB), both receiving a backspin ball. Kinematic parameters were measured using the MR3 myoMuscle Master Edition system (myoMOTION™, Noraxon, Scottsdale, AZ, USA, Figure 1). Noraxon′s inertial measurement units can be considered an alternative to the optical motion capture system for movement analysis. The IMU 3D angular measurement showed mostly good to high test–retest reliability with relatively small standard error of measurement [22]. During dynamic trials, the MSE (root mean squared error) for MyoMotion when compared with Vicon Motion Capture System (Vicon, Centennial, USA) is expected to be 0.50 and the correlation coefficient between Vicon and MyoMotion for dynamic trials to be 0.99 [23]. The accuracy and validity of the IMU system in angle determination is unquestioned and was a subject of previous research [24]. The myoMOTION system consists of a set of (1 to 16) sensors using inertial sensor technology. Based on so-called fusion algorithms, the information from a 3D accelerometer, gyroscope, and magnetometer is used to measure the 3D rotation angles of each sensor in absolute space (yaw–pitch–roll, also called orientation or navigation angles). Inertial sensors were located on the body of the study participant to record the accelerations, according to the myoMotion protocol, described in the manual (Figure 2).

Sensors were attached by the same technician with special straps and elastic self-adhesive tape. Every strap had its pocket for the inertial sensor. The straps with the sensors were light and easy to use and wear. The sensors were placed symmetrically so that the positive x-coordinate on the sensor label corresponded to a superior orientation for the trunk, head, and pelvis (Figure 2). Every participant, at the beginning of the measure, was checked and the system was calibrated. For the limb segment sensors, the positive x-coordinate corresponded to a proximal orientation. For the foot sensor, the x-coordinate was directed distally (to the toes). The sensors were placed according to the myoMotion manual protocol. The max sampling rate for a given sensor/receiver was 100 Hz per sensor for the whole 16-sensor set and was adjusted to the speed of registration by the piezoelectric sensor (1500 Hz).

Prior to the tasks, every participant followed standardized warm-up procedures: general (15 min) and table tennis-specific (20 min). Each task was composed of 15 specified strokes, and the player was asked to hit the marked area in the corner of the table (30 × 30 cm) diagonally (instruction given: “play diagonally, accurately and as hard as you can.” Every successful shot considered “on the table” and played diagonally was recorded for further analysis (missed balls, balls hit out of bounds, balls hit into the net, etc. were excluded). The balls were shot by a dedicated table tennis robot (Nevgy Robo Pong Robot 2050, Nevgy Industries, Hendersonville, TN, US, Figure 1) at constant parameters of rotation, speed, direction, and flight trajectory. The parameters of the robot were as follows: rotation given by robot–backspin (in both tasks),speed (determines both speed and spin, where 0 is the minimum and 30 is the maximum)–11 (in both tasks),left position (leftmost position to which the ball is delivered)–task 1: 4, task 2: 15,wing (robot’s head angle indicator)–9.5 (in both tasks),frequency (time interval between balls thrown)–1.4 s (in both tasks).

For the experiment, the same racket with the following characteristics was used: blade–Jonyer-H-AN (Butterfly, Tokyo, Japan), rubbers (both sides–Tenergy 05, 2.1 mm (Butterfly). Plastic Andro Speedball 3S 40 + balls (Andro, Dortmund, Germany) and a Stiga Premium Compact table (Stiga, Eskilstuna, Sweden) were used.

The following angles were recorded: knees flexion, hips flexion, hips abduction, hips rotation, lumbar rotation, lumbar flexion, lateral lumbar bending, chest rotation, chest flexion, lateral chest bending, playing-hand shoulder: flexion, abduction and rotation, playing-hand elbow flexion, playing-hand wrist: extension, supination and radial abduction. The maximum values of acceleration of the playing hand were also measured (ACCMax). The movement of the playing hand was used to assess specific events of the cycle: ready position (ready): hand not moving after the previous stroke, before the swing), backswing (backswing): the moment when the hand changes direction from backward to forward in the sagittal plane after the swing, and forward swing (forward): the moment when the hand changes direction from forward to backward in the sagittal plane after the stroke. The fourth event was defined by the moment when the maximum acceleration of playing hand was reached (max).

Statistical calculations were performed using Statistica 13.1 (TIBCO Software Inc., Palo Alto, CA, USA). The Shapiro-Wilk test was used to test the normality of data distribution for each variable on both sides (angular parameters across the four events and ACCMax) in each test group. The basic statistics were analysed (means, standard deviations–SD, and confidence intervals–CI–95% of all measured parameters). The Shapiro-Wilk test was used to assess the normality of the data distributions. The Mann-Whitney U test was used and Cohen′s d values were calculated to assess the differences between men and women. In addition, tests of significance examined whether these relationships were statistically significant. The results of the Mann-Whitney U test were considered significant for *p*
≤ 0.05, and Cohen′s d effect size interpretation was as follows: 20 ≤d<50—small effect size; 50 ≤d<80—medium effect size, d ≥ 80—large effect size [25].

## 3. Results

The parameters measured in the research are presented in Table 1 and Table 2.

In the ready event, statistically significant differences for topspin forehand were confirmed in the lumbar region position: a greater anterior pelvic tilt (ca. 5 degrees) and flexion towards the playing limb (about 4 degrees) was observed in men. The thoracic region in men and women was positioned similarly (with a significant difference in the flexion towards the playing limb, which was greater in women, see Table 1). The left hip in women had a higher level of limb abduction as compared to men (almost 8 degrees). In this event, the position of the playing arm was different in men than in women; the hand in men was more extended (ca. 11 degrees) and adducted (ca. 26 degrees) at the shoulder joint, extended at the elbow joint (ca. 40 degrees) and at the wrist joint (2 degrees). At the same time, men tended to hold the hand in the wrist joint in lower abduction (ca. 15 degrees) and pronation (ca. 16 degrees) than women. In the backswing event, men showed greater shoulder extension, abduction, pronation, elbow extension, and greater right hip flexion and abduction compared to women. The angle of wrist extension and supination was also greater in men than in women. In the max event, the differences related to the left hip joint, with men showing greater flexion (ca. 14 degrees) and adduction (ca. 9 degrees). Greater pronation in men than in women also occurred in the shoulder joint (almost 8 degrees) and the right hip (ca. 23 degrees). The knee flexion angle was also larger in men than in women (left–ca. 7, right–over 12 degrees). In the Forward event, the differences between men and women pertained to greater thoracic extension and flexion towards the non-playing limb (both values of several degrees), greater left hip flexion and pronation (ca. 11 and 15 degrees, respectively), greater knee flexion (left–5 degrees, right–10 degrees), greater wrist flexion, and pronation and abduction towards the radial bone (12, 18 and 42 degrees, respectively) in men. Women showed a greater angle of elbow flexion (over 40 degrees) and shoulder pronation (almost 10 degrees). The above-mentioned differences were statistically significant, as evidenced by the *p*-value of the Mann-Whitney U test and the Cohen’s d effect size. The range of motion in the elbow joint (backswing to forward), 10 degrees greater in women than in men, is worth noting, as is the range of wrist flexion. At the shoulder joint, on the other hand, the flexion movement is about 20 degrees greater in men than in women (Table 1). The value of the maximum hand acceleration is much higher in men than in women, by almost 50 m/s^2^.

In the topspin backhand shot (Table 2), in the ready event, the differences were mainly observed in the position of the playing limb at individual joints; men had greater shoulder pronation and adduction, lower elbow and wrist flexion, and greater abduction of the hand towards the radial bone in this joint as compared to women. In the backswing event, greater chest rotation and flexion towards the non-playing limb in men compared to women (several degree differences), shoulder abduction, and greater differences in the position of the wrist joint (greater in women: pronation, elbow flexion, and extension) were found. There was also greater supination (ca. 20 degrees), right hip flexion (ca. 8 degrees), left hip pronation (ca. 10 degrees), and right knee flexion (ca. 15 degrees) in men as compared to women. In the max event, the differences concerned the thoracic region, left hip (for flexion and supination, the differences reached 20 degrees), and the playing limb.

A greater value of pronation was observed in the shoulder joint (30 degrees) and in the elbow joint, with greater flexion angle (ca. 15 degrees) in men. There were also some differences between the two groups in the wrist (supination greater in women by ca. 18 degrees) and the left hip (in men, flexion was greater by over 20 degrees and abduction by 7 degrees, whereas supination was greater in women by ca. 18 degrees). In the right hip, flexion was greater in men by almost 15 degrees, while supination and abduction were greater in women. There was also a larger knee flexion in men: in the left knee by ca. 18 degrees and in the right by ca. 30 degrees. Similar differences in the range were observed in the forward phase. A greater range of the elbow extension (ca. 7 degrees) from backswing to forward and left knee extension, shoulder pronation, and wrist supination, observed in women compared to men, is also noticeable. In men, a greater range of shoulder flexion as compared to women was observed from backswing to forward. The maximum hand acceleration in men was higher than in women (ca. 20 m/s^2^).

## 4. Discussion

The aim of this study was to evaluate the differences in the values of selected angular and kinematic parameters between female and male table tennis players during topspin forehand and topspin backhand shots played against the backspin ball. To the best of our knowledge, this subject has not yet been addressed in existing literature. Comparative research in table tennis has most often concerned players of different levels, age, training experience, etc. [21,26,27] or comparison of the kinematics of different shots [28,29]. Gender differences in kinematics have been found in other sports, emphasizing the need to individualize and adapt training programs to the gender of athletes [15,30,31].

The results of the tests performed in the present study demonstrated the occurrence of many gender differences in the values of angular parameters in both shots. This demonstrates the differences in the performance of these techniques by men and women. Changes in the angular parameters and ranges of motion from backswing through max to forward events show that, in men, the topspin forehand shot is supported by greater involvement of the knee and hip, chest rotation and flexion, shoulder flexion, and abduction. In women, greater use of elbow movements and shoulder pronation was observed. Perhaps these differences were due to the different anatomical structures and, thus, the different potential of the biomechanics of movement in men and women. The big muscle masses of the trunk, hip, and shoulder girdle in men perhaps give them the opportunity to generate a higher force than in women. It can be assumed that that, in men, there is greater movement of the trunk, knees, hips and shoulder (especially flexion) as compared to women, which leads to the difference in maximum acceleration between women and men. It can be therefore presumed that technique training in women should take these differences into account. However, the “male model” of making a topspin forehand shot may not necessarily be better for women. As argued by Serrien et al., the differences in the kinematics of handball throwing movements between men and women may be due to the different sizes and anthropometric profiles of female and male players [32]. Differentiation of movement patterns can therefore be a manifestation of movement optimization stemming from anthropological differences and limitations. Another important aspect is the awareness of possible different susceptibilities to injuries between women and men, which has also been noted in other sports [33,34].

In the topspin backhand shot, it was observed that men, more than women, use (in the main forward phase) the shoulder flexion movement and supporting movements in the hip and chest rotation. Perhaps similar to the topspin forehand, men generate higher power from big muscles of the trunk, shoulder and hip girdle by rotational movement of the body and shoulder flexion. Furthermore, women are characterized by greater use of shoulder supination, elbow extension, and wrist supination during the impact phase. It can be presumed that the pattern of the impact movements includes movements that make less use of large muscle groups and large joints (hip joints, chest joints, and shoulder joint in extension and flexion movements) and more use of small muscle groups and small joints (elbow joints, wrist joints) than men. The use of supination at shoulder joint, extension at elbow and supination at wrist in females could characterize more effective use of playing hand, owing to less power coming from the body than in topspin backhand in men.

The differences observed in the maximum hand acceleration values suggest that the potential of using topspin forehand to perform a strong aggressive play with more force compared to topspin backhand may be typical for men. Women were characterized by a slightly greater acceleration of maximum topspin backhand than forehand, which suggests a slightly different use of these two shots as compared to men. It is likely that women could use both sides to perform a topspin attack against the backspin ball, while men could seek opportunities to make a stronger shot with a forehand topspin. Differences in the use of shots between men and women have been already demonstrated in table tennis [35]. Gender differences in tactical solutions were also noted in badminton and tennis [36,37].

As limitations of our study, it should be also mentioned that the groups compared, although constituting national elite table tennis players, do not include the world leaders. These observations should be confirmed by conducting similar tests on different world elite male and female table tennis players. The tests also concerned individual shots performed under relatively constant and reproducible conditions. They did not require constant evaluation of the ball parameters and their adjustment. Perhaps the tests in the varying playing conditions would yield different findings.

## 5. Conclusions

The differences found in the magnitude of angular parameters and maximum hand acceleration between men and women are probably a manifestation of gender differentiation of movement patterns of topspin forehand and topspin backhand. It can be assumed that women benefit from the movements of small muscle groups and small joints (elbow and wrist joints) during topspin shots to a greater extent than men. The movement pattern of topspin strokes performed by men takes into account, more than that in the case of women, movements that use large muscle groups and large joints (hip joints, trunk joints, and shoulder joints in extension and flexion). Differentiation of movement patterns can be a manifestation of movement optimization due to anthropological differences and limitations. The big muscles mass of the trunk, hip, and shoulder girdle of men perhaps give them the opportunity to generate a higher force during topspin forehand and backhand than in women. In addition, the use of supination at the shoulder joint, extension at elbow, and supination at wrist in females characterize more effective use of the playing hand, owing to less power coming from the body than in topspin of males. The differences in the values of maximal acceleration suggest that women could use both sides to perform a topspin attack against the backspin ball, while men could seek opportunities to make a stronger shot with a forehand topspin.

## Figures and Tables

**Figure 1 ijerph-17-05742-f001:**
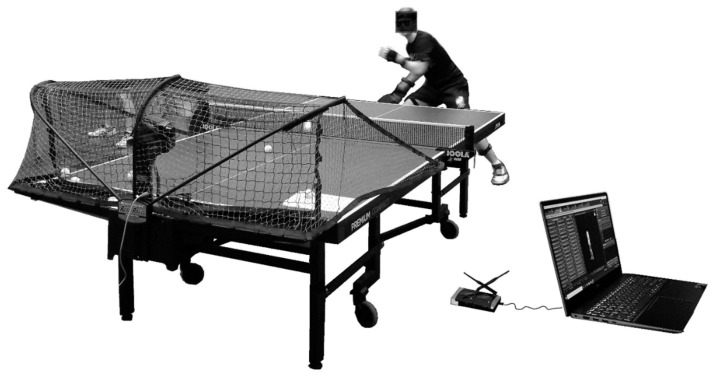
Experimental design.

**Figure 2 ijerph-17-05742-f002:**
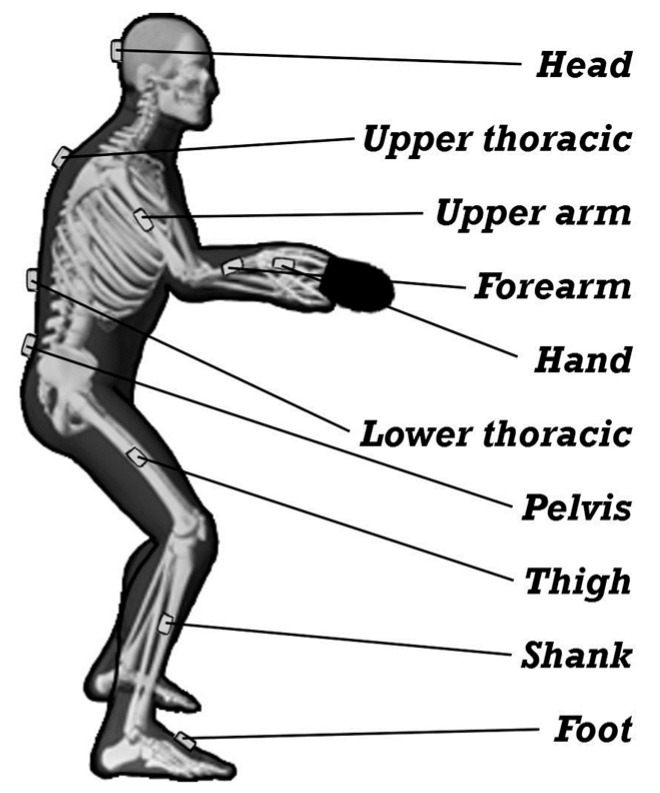
Sensor location.

**Table 1 ijerph-17-05742-t001:** The values of the parameters of topspin forehand of women (*n* = 6) and men (*n* = 6).

	Mean ± SD (CI 95%)		*p*	Cohen’s d
	Men	Women		
**Ready Position**				
LumbarRotation(deg)	−4.38 ± 12.53 (−6.89 ÷ −1.86)	−1.38 ± 2.68 (−1.96 ÷ −0.80)	0.57	0.39
LumbarFlexion(deg)	21.65 ± 8.65 (19.91 ÷ 23.38)	16.47 ± 2.86 (15.85÷17.09)	<0.01	0.90 **
LumbarLateral(deg)	3.33 ± 3.87 (2.55 ÷ 4.10)	−1.12 ± 8.04 (−2.87 ÷ 0.63	<0.01	0.75 *
ThoracicRotation(deg)	−2.25 ± 8.47 (3.95 ÷ −0.55)	−2.79 ± 8.80 (−4.70 ÷ −0.88)	0.87	0.06
ThoracicFlexion(deg)	−7.79 ± 6.51 (−9.10 ÷ −6.49)	−9.71 ± 8.45 (−11.54÷ −7.87)	0.24	0.26
ThoracicLateral(deg)	2.00 ± 5.08 (0.98 ÷ 3.02)	5.28 ± 7.94 (3.56 ÷ 7.01)	<0.01	0.5 *
HipLTFlexion(deg)	39.87 ± 12.83 (37.30 ÷ 42.44)	42.27 ± 9.20 (40.27 ÷ 44.27)	0.65	0.22
HipLTAbduction(deg)	19.08 ± 17.56 (15.56 ÷ 22.60)	26.86 ± 5.57 (25.65÷28.07)	<0.01	0.67 *
HipLTRotationExt(deg)	−0.57 ± 20.30 (−4.64 ÷ 3.50)	5.92 ± 9.53 (3.85 ÷ 7.99)	<0.01	0.44
HipRTFlexion(deg)	35.98 ± 20.53 (31.87 ÷ 40.10)	35.34 ± 13.98 (32.31 ÷ 38.38)	0.38	0.04
HipRTAbduction(deg)	17.93 ± 14.56 (15.01 ÷ 20.85)	22.61 ± 5.08 (21.51 ÷ 23.71)	0.28	0.48
HipRTRotationExt(deg)	−0.19 ± 23.77 (−4.96 ÷ 4.57)	13.90 ± 10.99 (11.52 ÷ 16.29)	<0.01	0.81 **
KneeLTFlexion(deg)	43.04 ± 8.19 (41.40 ÷ 44.68)	41.16 ± 10.93 (38.79 ÷ 43.53)	0.1	0.2
KneeRTFlexion(deg)	39.66 ± 11.87 (37.28 ÷ 42.04)	41.24 ± 10.99 (38.85 ÷ 43.63)	0.29	0.14
ShoulderRTRotationExt(deg)	−33.31 ± 41.01 (−41.53 ÷ −25.08)	−36.15 ± 40.01 (−44.83 ÷ 27.47)	0.23	0.07
ShoulderRTFlexion(deg)	23.61 ± 15.61 (20.48 ÷ 26.74)	34.24 ± 17.71 (30.40 ÷ 38.09)	<0.01	0.64 *
ShoulderRTAbduction(deg)	1.31 ± 13.74(−1.44 ÷ 4.07)	27.69 ± 27.70 (21.68 ÷ 33.70)	<0.01	1.27 **
ElbowRTFlexion(deg)	49.66 ± 45.98 (40.44 ÷ 58.87)	90.23 ± 13.99 (67.19 ÷ 93.27)	<0.01	1.35 **
WristRTExtension(deg)	31.53 ± 38.01 (23.91 ÷ 39.15)	29.68 ± 47.45 (19.38 ÷3 9.98)	0.03	0.04
WristRTRadial(deg)	1.64 ± 13.42 (−1.05 ÷ 4.34)	−17.02 ± 19.01 (−21.15 ÷ −12.90)	<0.01	1.15 **
WristRTSupination(deg)	11.31 ± 17.42 (7.82 ÷ 14.81)	27.57 ± 25.72 (21.99 ÷ 33.15)	<0.01	0.75 *
**Backswing Position**				
LumbarRotation(deg)	−2.14 ± 12.20 (−4.59 ÷ 0.30)	2.59 ± 2.63 (2.02 ÷ 3.16)	0.01	0.64 *
LumbarFlexion(deg)	25.72 ± 8.37 (24.04 ÷ 27.40)	22.30 ± 5.92 (21.01 ÷ 23.58)	<0.01	0.48
LumbarLateral(deg)	5.25 ± 8.56 (3.53 ÷ 6.97)	−2.74 ± 9.18 (−4.73 ÷ −0.74)	<0.01	0.90 **
ThoracicRotation(deg)	3.34 ± 9.57 (1.42 ÷ 5.26)	−5.58 ± 7.90 (−7.29 ÷ −3.86)	<0.01	1.02 **
ThoracicFlexion(deg)	−11.68 ± 9.22 (012.53 ÷ −9.83)	−9.77 ± 13.32 (−12.66 ÷ 6.88)	0.22	0.17
ThoracicLateral(deg)	11.80 ± 14.80 (8.83 ÷ 14.76)	9.58 ± 14.39 (6.45 ÷ 12.70)	0.06	0.15
HipLTFlexion(deg)	22.35 ± 10.19 (20.31 ÷ 24.40)	35.18 ± 30.10 (28.65 ÷ 41.71)	0.19	0.64 *
HipLTAbduction(deg)	21.12 ± 13.61 (18.39 ÷ 23.85)	23.91 ± 5.59 (22.70 ÷ 25.13)	0.14	0.29
HipLTRotationExt(deg)	17.92 ± 15.28 (14.86 ÷ 20.99)	18.53 ± 23.12 (13.51 ÷ 23.55)	0.19	0.03
HipRTFlexion(deg)	79.78 ± 13.77 (77.02 ÷ 82.54)	65.45 ± 16.99 (61.76 ÷ 69.14)	<0.01	0.93 **
HipRTAbduction(deg)	0.08 ± 19.11 (−3.76 ÷ 3.91)	9.47 ± 13.79 (6.48 ÷ 12.45)	<0.01	0.57 *
HipRTRotationExt(deg)	−28.86 ± 13.00 (−31.47 ÷ −26.25)	−6.73 ± 13.40(−9.64÷ −3.83)	<0.01	1.68 **
KneeLTFlexion(deg)	65.33 ± 11.36 (63.05 ÷ 67.61)	61.85 ± 13.11 (59.01 ÷ 64.70)	0.16	0.28
KneeRTFlexion(deg)	64.16 ± 13.92(61.37 ÷ 66.95)	57.98 ± 11.21 (55.54 ÷ 60.410	<0.01	0.49
ShoulderRTRotationExt(deg)	−10.57 ± 42.21 (−19.03 ÷ −2.11)	−18.09 ± 46.78 (−28.24 ÷ −7.94)	0.98	0.17
ShoulderRTFlexion(deg)	−1.63 ± 29.15 (−7.45 ÷ 4.22)	17.04 ± 40.06 (8.35 ÷ 25.73)	<0.01	0.54 *
ShoulderRTAbduction(deg)	38.97 ± 30.01 (32.95 ÷ 44.99)	36.26 ± 32.72 (29.16 ÷ 43.36)	0.05	0.09
ElbowRTFlexion(deg)	27.33 ± 24.32 (22.45 ÷ 32.20)	46.83 ± 36.81 (38.84 ÷ 54.81)	<0.01	0.64 *
WristRTExtension(deg)	15.54 ± 22.32 (11.06 ÷ 20.01)	3.83 ± 20.08 (−0.53 ÷ 8.19)	<0.01	0.55 *
WristRTRadial(deg)	−15.78 ± 15.61(−18.91÷ −12.65)	−18.07 ± 19.40(−22.28÷ −13.86)	0.19	0.13
WristRTSupination(deg)	49.38 ± 72.40(34.87÷63.90)	21.38 ± 21.74(16.66÷26.10)	0.21	0.5 *
**Maximal acceleration**				
LumbarRotation(deg)	−7.44 ± 12.80(−10.03 ÷ −4.85)	−2.64 ± 4.89(−3.70÷ −1.57)	0.1	0.54 *
LumbarFlexion(deg)	15.27 ± 10.13(13.21 ÷ 17.32)	12.69 ± 4.96(11.61 ÷ 13.78)	0.12	0.34
LumbarLateral(deg)	3.98 ± 3.52(3.27 ÷ 4.69)	0.72 ± 9.32(−1.32 ÷ 2.75)	0.5	0.51 *
ThoracicRotation(deg)	5.48 ± 7.86(3.89 ÷ 7.08)	5.63 ± 12.85(2.82 ÷ 8.43)	0.29	0.01
ThoracicFlexion(deg)	−9.71 ± 6.95(−11.12 ÷ −8.30)	−4.01 ± 8.55(−5.88 ÷ −2.15)	<0.01	0.73 *
ThoracicLateral(deg)	−0.27 ± 7.00(−1.69 ÷ 1.15)	−0.30 ± 10.73(−2.65 ÷ 2.04)	0.73	0.00
HipLTFlexion(deg)	41.96 ± 15.50(38.82 ÷ 45.10)	28.37 ± 15.16(24.84 ÷ 31.90)	<0.01	0.86 **
HipLTAbduction(deg)	27.51 ± 15.39(24.39 ÷ 30.63)	30.80 ± 11.20(28.36 ÷ 33.25)	0.04	0.25
HipLTRotationExt(deg)	−8.50 ± 16.40(−11.82 ÷ −5.17)	0.79 ± 11.17(−1.65 ÷ 3.23)	<0.01	0.67 *
HipRTFlexion(deg)	39.14 ± 14.22(36.25 ÷ 42.02)	32.19 ± 21.89(27.41 ÷ 36.96)	<0.01	0.38
HipRTAbduction(deg)	18.17 ± 19.83(14.15 ÷ 22.19)	26.71 ± 9.44(24.65 ÷ 28.77)	0.01	0.58 *
HipRTRotationExt(deg)	−6.64 ± 15.97(−9.87 ÷ −3.40)	16.27 ± 16.34(12.70 ÷ 19.84)	<0.01	1.42 **
KneeLTFlexion(deg)	55.21 ± 10.06(53.17 ÷ 57.24)	48.93 ± 14.45(45.77 ÷ 52.08)	<0.01	0.51 *
KneeRTFlexion(deg)	67.31 ± 9.64(65.35 ÷ 69.26)	55.08 ± 15.69(51.66 ÷ 58.51)	<0.01	0.97 **
ShoulderRTRotationExt(deg)	−30.08 ± 38.57(−37.89 ÷ −22.26)	−22.59 ± 40.93(−31.53 ÷ −13.66)	<0.01	0.19
ShoulderRTFlexion(deg)	52.75 ± 32.48(46.17 ÷ 59.34)	55.67 ± 21.81(50.91 ÷ 60.43)	0.2	0.11
ShoulderRTAbduction(deg)	29.68 ± 40.27(21.52 ÷ 37.84)	32.71 ± 37.69(24.48 ÷ 40.94)	0.59	0.08
ElbowRTFlexion(deg)	30.64 ± 81.55(14.10 ÷ 47.19)	62.81 ± 26.97(56.92 ÷ 68.70)	0.01	0.59 *
WristRTExtension(deg)	28.81 ± 33.41(22.04 ÷ 35.58)	38.91 ± 38.69(30.47 ÷ 47.36)	0.18	0.28
WristRTRadial(deg)	−14.78 ± 31.02(−21.07 ÷ −8.50)	−13.60 ± 20.56(−18.09 ÷ −9.11)	0.8	0.05
WristRTSupination(deg)	22.87 ± 51.76(12.39 ÷ 33.36)	19.04 ± 30.59(12.39 ÷ 25.72)	0.96	0.09
**Forward Position**				
LumbarRotation(deg)	−7.06 ± 13.16(−9.70 −4.42)	−3.36 ± 4.06(−4.22 ÷ −2.50)	<0.01	0.43
LumbarFlexion(deg)	14.37 ± 11.72(12.02 ÷ 16.73)	11.45 ± 5.12(10.37 ÷ 12.54)	0.4	0.35
LumbarLateral(deg)	1.68 ± 2.46(1.18 ÷ 2.17)	−1.29 ± 7.74(−2.93 ÷ 0.35)	0.2	0.58 *
ThoracicRotation(deg)	5.12 ± 6.61(3.80 ÷ 6.45)	4.40 ± 14.14(1.40 ÷ 7.40)	0.2	0.07
ThoracicFlexion(deg)	−6.80 ± 7.14(−8.23 ÷ −5.37)	−2.65 ± 5.95(−3.91 ÷ −1.39)	0.5	0.63 *
ThoracicLateral(deg)	−1.55 ± 3.75(−2.30 ÷ 0.80)	3.32 ± 10.10(1.18 ÷ 5.46)	<0.01	0.7 *
HipLTFlexion(deg)	46.88 ± 16.48(43.58 ÷ 50.19)	37.92 ± 9.53(35.90 ÷ 39.94)	<0.01	0.69 *
HipLTAbduction(deg)	15.21 ± 19.52(11.29 ÷ 19.12)	23.83 ± 12.51(21.18 ÷ 26.49)	<0.01	0.54
HipLTRotationExt(deg)	−11.20 ± 14.07(−14.02 ÷ −8.38)	3.21 ± 6.87(1.75 ÷ 4.67)	<0.01	1.38 **
HipRTFlexion(deg)	19.69 ± 18.49(15.99 ÷ 23.40)	25.58 ± 28.44(19.56 ÷ 31.61)	<0.01	0.25
HipRTAbduction(deg)	19.74 ± 12.25(17.28 ÷ 22.19)	25.78 ± 8.50(23.98 ÷ 27.58)	0.76	0.58 *
HipRTRotationExt(deg)	13.20 ± 20.14(9.16 ÷ 17.24)	23.30 ± 19.14(19.24 ÷ 27.35)	<0.01	0.51 *
KneeLTFlexion(deg)	44.76 ± 10.90(42.58 ÷ 46.95)	39.13 ± 18.22(35.27 ÷ 42.96)	<0.01	0.39
KneeRTFlexion(deg)	56.64 ± 15.09(53.61 ÷ 59.66)	46.64 ± 16.47(43.15 ÷ 50.13)	<0.01	0.63 *
ShoulderRTRotationExt(deg)	−42.25 ± 42.75(−50.82 ÷ −33.68)	−51.51 ± 35.90(−59.12 ÷ −43.90)	<0.01	0.24
ShoulderRTFlexion(deg)	102.36 ± 54.73(91.39 ÷ 113.34)	96.10 ± 27.85(90.20 ÷ 102.01)	0.16	0.15
ShoulderRTAbduction(deg)	110.70 ± 101.55(90.34 ÷ 131.05)	111.44 ± 92.83(91.77 ÷ 131.11)	0.05	0.01
ElbowRTFlexion(deg)	39.99 ± 66.92(26.57 ÷ 53.40)	77.30 ± 20.57(72.94 ÷ 81.66)	0.79	0.85 **
WristRTExtension(deg)	16.03 ± 27.30(10.56 ÷ 21.51)	4.77 ± 19.85(0.56 ÷ 8.97)	<0.01	0.48
WristRTRadial(deg)	−24.68 ± 30.44(−30.82 ÷ −18.55)	−6.44 ± 24.77(−11.69 ÷ −1.19)	0.09	0.65 *
WristRTSupination(deg)	−0.39 ± 66.69(−13.76 ÷ 12.98)	42.55 ± 93.48(22.75 ÷ 62.36)	<0.01	0.54 *
ACCMax(m/s^2^)	210.69 ± 15.77(207.49 ÷ 213.89)	164.69 ± 18.15(160.78 ÷ 168.61)	<0.01	2.71 **

Data are presented as mean ± SD (Confidence intervals). Abbreviations: ACCMax—maximal acceleration of the playing hand, Ext—external, LT—left, RT—right, *p*—value of p U Mann−Whitney test, p and Cohen’s d statistical significances: *—medium effect size, **—large effect size.

**Table 2 ijerph-17-05742-t002:** The values of the parameters of topspin backhand of women (*n* = 6) and men (*n* = 6).

	Mean ± SD (CI 95%)		*p*	Cohen’s d
	Men	Women		
**Ready Position**				
LumbarRotation(deg)	−3.86 ± 14.43(−6.84 ÷ −0.87)	−0.34 ± 6.08(−1.67 ÷ 0.99)	0.01	0.34
LumbarFlexion(deg)	20.70 ± 7.84(19.07 ÷ 22.34)	15.45 ± 4.48(14.47 ÷ 16.43)	<0.01	0.85 **
LumbarLateral(deg)	0.97 ± 3.84(0.17 ÷ 1.77)	−2.55 ± 6.40(−3.95 ÷ −1.15)	<0.01	0.69 *
ThoracicRotation(deg)	−1.55 ± 6.76(−2.96 ÷ 0.14)	−0.20 ± 3.53(−0.97 ÷ 0.57)	0.04	0.26
ThoracicFlexion(deg)	−4.35 ± 5.65(−5.53 ÷ −3.18)	−3.87 ± 8.77(−5.78 ÷ −1.95)	0.61	0.07
ThoracicLateral(deg)	−1.95 ± 3.11(−2.60 ÷ −1.30)	5.04 ± 8.76(3.13 ÷ 6.95)	<0.01	1.18 **
HipLTFlexion(deg)	32.98 ± 18.81(29.07 ÷ 36.90)	34.51 ± 14.79(31.28 ÷ 37.74)	0.94	0.09
HipLTAbduction(deg)	28.75 ± 10.51(26.56 ÷ 30.94)	28.72 ± 6.47(27.31 ÷ 30.14)	0.76	0.00
HipLTRotationExt(deg)	5.86 ± 18.85(1.93 ÷ 9.79)	3.56 ± 11.52(1.04 ÷ 6.08)	0.4	0.15
HipRTFlexion(deg)	43.65 ± 14.58(40.62 ÷ 46.69)	40.26 ± 10.24(38.03 ÷ 42.50)	0.01	0.27
HipRTAbduction(deg)	17.06 ± 15.19(13.89 ÷ 20.22)	21.00 ± 5.46(19.81 ÷ 22.20)	0.64	0.38
HipRTRotationExt(deg)	−9.26 ± 19.05(−13.23 ÷ −5.30)	4.77 ± 9.03(2.80 ÷ 6.74)	<0.01	1.00 **
KneeLTFlexion(deg)	39.25 ± 10.78(37.01 ÷ 41.50)	43.05 ± 9.91(40.88 ÷ 45.21)	0.03	0.37
KneeRTFlexion(deg)	42.15 ± 9.82(40.10 ÷ 44.19)	36.97 ± 10.43(34.69 ÷ 39.24)	0.01	0.51 *
ShoulderRTRotationExt(deg)	−37.29 ± 27.39(−43 ÷ −31.59)	−10.18 ± 19.29(−14.39 ÷ −5.97)	<0.01	1.16 **
ShoulderRTFlexion(deg)	14.22 ± 15.08(11.08 ÷ 17.36)	17.63 ± 16.62(14.00 ÷ 21.26)	0.11	0.21
ShoulderRTAbduction(deg)	18.12 ± 16.65(14.65 ÷ 21.59)	38.36 ± 19.33(34.14 ÷ 42.58)	<0.01	1.12 **
ElbowRTFlexion(deg)	71.71 ± 13.03(69.00 ÷ 74.42)	85.39 ± 16.72(81.74 ÷ 89.05)	<0.01	0.92 **
WristRTExtension(deg)	−2.33 ± 16.48(−5.76 ÷ 1.11)	−25.44 ± 21.19(−30.07 ÷ −20.82)	<0.01	1.23 **
WristRTRadial(deg)	12.18 ± 13.26(9.42 ÷ 14.94)	−3.33 ± 12.16(−5.99 ÷ −0.68)	<0.01	1.22 **
WristRTSupination(deg)	36.23 ± 21.07(31.84 ÷ 40.62)	43.62 ± 19.77(39.30 ÷ 47.94)	0.02	0.36
**Backswing Position**				
LumbarRotation(deg)	−5.77 ± 12.25(−8.32 ÷ −3.22)	−0.91 ± 6.45(−2.32 ÷ 0.50)	0.02	0.52 *
LumbarFlexion(deg)	25.39 ± 8.22(23.68 ÷ 27.11)	19.62 ± 3.46(18.86 ÷ 20.37)	<0.01	0.99 **
LumbarLateral(deg)	−0.64 ± 7.66(−2.24 ÷ 0.95)	−4.30 ± 6.68(−5.76 ÷ −2.84)	<0.01	0.51 *
ThoracicRotation(deg)	−8.49 ± 7.67(−10.09 ÷ −6.89)	−3.48 ± 8.37(−5.30 ÷ −1.65)	<0.01	0.62 *
ThoracicFlexion(deg)	−3.05 ± 10.22(−5.18 ÷ −0.92)	−2.80 ± 8.48(−4.65 ÷ −0.95)	0.63	0.03
ThoracicLateral(deg)	−5.00 ± 10.60(−7.21 ÷ −2.80)	3.14 ± 10.90(0.76 ÷ 5.52)	<0.01	0.76 *
HipLTFlexion(deg)	66.51 ± 21.23(62.08 ÷ 70.93)	59.09 ± 11.37(56.61 ÷ 61.58)	<0.01	0.45
HipLTAbduction(deg)	29.21 ± 10.20(27.09 ÷ 31.34)	29.18 ± 7.68(27.50 ÷ 30.85)	0.73	0.00
HipLTRotationExt(deg)	−16.53 ± 12.12(−19.06 ÷ −14.00)	−6.57 ± 9.51(−8.64 ÷ −4.49)	<0.01	0.92 **
HipRTFlexion(deg)	64.84 ± 15.23(61.67 ÷ 68.02)	56.86 ± 15.88(53.39 ÷ 60.32)	<0.01	0.51 *
HipRTAbduction(deg)	21.05 ± 18.69(17.16 ÷ 24.94)	29.41 ± 7.20(27.84 ÷ 30.99)	0.01	0.65 *
HipRTRotationExt(deg)	−11.63 ± 19.79(−15.75 ÷ −7.51)	8.13 ± 15.58(4.73 ÷ 11.53)	<0.01	1.12 **
KneeLTFlexion(deg)	51.71 ± 13.54(48.89 ÷ 54.53)	47.27 ± 19.08(43.11 ÷ 51.44)	0.09	0.27
KneeRTFlexion(deg)	74.12 ± 11.27(71.77 ÷ 76.46)	59.82 ± 7.18(58.25 ÷ 61.39)	<0.01	1.55 **
ShoulderRTRotationExt(deg)	−59.02 ± 32.84(−65.86 ÷ −52.18)	−62.82 ± 37.37(−70.98 ÷ −54.66)	0.26	0.11
ShoulderRTFlexion(deg)	19.23 ± 14.97(16.11 ÷ 22.35)	25.87 ± 28.75(19.59 ÷ 32.15)	0.56	0.3
ShoulderRTAbduction(deg)	22.91 ± 14.12(19.97 ÷ 25.85)	17.20 ± 29.97(10.66 ÷ 23.75)	0.01	0.26
ElbowRTFlexion(deg)	59.90 ± 12.35(57.33 ÷ 62.47)	59.36 ± 21.54(54.66 ÷ 64.06)	0.77	0.03
WristRTExtension(deg)	−12.98 ± 19.01(−16.94 ÷ −9.02)	−18.72 ± 14.22(−21.83 ÷ −15.52)	0.08	0.35
WristRTRadial(deg)	−1.51 ± 16.26(−4.89 ÷ 1.88)	−21.01 ± 16.55(−24.62 ÷ −17.40)	<0.01	1.19 **
WristRTSupination(deg)	33.35 ± 28.03(27.51 ÷ 39.19)	18.29 ± 26.96(12.40 ÷ 24.18)	<0.01	0.55 *
**Maximal Acceleration Position**				
LumbarRotation(deg)	−5.50 ± 12.28(−8.18 ÷ −2.82)	−2.68 ± 7.08(−4.23 ÷ −1.12)	0.7	0.28
LumbarFlexion(deg)	20.06 ± 8.90(18.20 ÷ 21.91)	12.74 ± 3.65(11.93 ÷ 13.54)	<0.01	1.17 **
LumbarLateral(deg)	−0.40 ± 4.15(−1.26 ÷ 0.47)	−1.96 ± 5.29(−3.13 ÷ −0.80)	0.1	0.33
ThoracicRotation(deg)	−0.67 ± 7.95(−2.32 ÷ 0.99)	−0.94 ± 4.82(−1.99 ÷ 0.12)	0.8	0.04
ThoracicFlexion(deg)	−1.48 ± 5.65(−2.66 ÷ −0.30)	3.11 ± 6.86(1.60 ÷ 4.62)	<0.01	0.73 *
ThoracicLateral(deg)	−10.81 ± 9.71(−12.84 ÷ −8.79)	−3.24 ± 16.23(−6.81 ÷ 0.33)	0.01	0.58 *
HipLTFlexion(deg)	48.25 ± 17.99(44.50 ÷ 51.99)	27.05 ± 16.44(23.43 ÷ 30.66)	<0.01	1.23 **
HipLTAbduction(deg)	33.05 ± 12.46(30.46 ÷ 35.65)	25.99 ± 5.96(24.68 ÷ 27.29)	<0.01	0.77 *
HipLTRotationExt(deg)	−12.70 ± 15.39(−15.90 ÷ −9.49)	6.94 ± 10.38(4.66 ÷ 9.22)	<0.01	1.52 **
HipRTFlexion(deg)	52.55 ± 15.61(49.30 ÷ 55.80)	36.40 ± 11.37(33.90 ÷ 38.90)	<0.01	1.20 **
HipRTAbduction(deg)	20.40 ± 18.99(16.44 ÷ 24.35)	27.36 ± 3.30(26.64 ÷ 28.09)	0.80	0.62 *
HipRTRotationExt(deg)	−10.52 ± 19.21(−14.52 ÷ −6.52)	10.11 ± 8.09(8.33 ÷ 11.89)	<0.01	1.51 **
KneeLTFlexion(deg)	47.07 ± 14.28(44.10 ÷ 50.05)	29.24 ± 18.62(25.15 ÷ 33.33)	<0.01	1.08 **
KneeRTFlexion(deg)	72.10 ± 11.71(69.66 ÷ 74.54)	42.64 ± 11.29(40.15 ÷ 45.12)	<0.01	2.56 **
ShoulderRTRotationExt(deg)	−73.02 ± 28.52(−78.96 ÷ −67.08)	−47.08 ± 27.32(−53.09 ÷ −41.08)	<0.01	0.93 **
ShoulderRTFlexion(deg)	70.57 ± 18.67(66.68 ÷ 74.46)	62.47 ± 244.92(57.00 ÷ 67.94)	0.02	0.37
ShoulderRTAbduction(deg)	63.49 ± 62.98(50.37 ÷ 76.60)	52.05 ± 31.71(45.09 ÷ 59.02)	0.41	0.24
ElbowRTFlexion(deg)	62.97 ± 22.05(58.38 ÷ 67.56)	48.91 ± 19.46(44.63 ÷ 53.19)	<0.01	0.68 *
WristRTExtension(deg)	−9.00 ± 27.06(−14.64 ÷ −3.36)	−19.92 ± 23.01(−24.98 ÷ −14.86)	0.01	0.44
WristRTRadial(deg)	−2.08 ± 21.70(−6.60 ÷ 2.44)	1.98 ± 11.41(−0.53 ÷ 4.49)	0.76	0.25
WristRTSupination(deg)	37.84 ± 28.29(31.95 ÷ 43.74)	55.85 ± 24.16(50.54 ÷ 61.16)	<0.01	0.69 *
**Forward Position**				
LumbarRotation(deg)	−5.50 ± 12.28(−8.18 ÷ −2.82)	−2.68 ± 7.08(−4.23 ÷ −1.12)	0.7	0.28
LumbarFlexion(deg)	20.06 ± 8.90(18.20 ÷ 21.91)	12.74 ± 3.65(11.93 ÷ 13.54)	<0.01	0.97 **
LumbarLateral(deg)	−0.40 ± 4.15(−1.26 ÷ 0.47)	−1.96 ± 5.29(−3.13 ÷ −0.80)	0.1	0.83 **
ThoracicRotation(deg)	−0.67 ± 7.95(−2.32 ÷ 0.99)	−0.94 ± 4.82(−1.99 ÷ 0.12)	0.8	0.56 *
ThoracicFlexion(deg)	−2.33 ± 5.79(−3.50 ÷ −1.15)	2.24 ± 8.77(0.37 ÷ 4.11)	<0.01	0.63 *
ThoracicLateral(deg)	−4.60 ± 7.75(−6.18 ÷ −3.02)	−1.48 ± 14.21(−4.51 ÷ 1.55)	0.92	0.28
HipLTFlexion(deg)	36.07 ± 22.38(31.51 ÷ 40.63)	23.98 ± 16.50(20.47 ÷ 27.50)	<0.01	0.62 *
HipLTAbduction(deg)	31.96 ± 12.56(29.40 ÷ 34.51)	26.48 ± 6.44(25.11 ÷ 27.86)	<0.01	0.58 *
HipLTRotationExt(deg)	−2.73 ± 22.48(−7.31 ÷ 1.85)	6.61 ± 10.06(4.46 ÷ 8.75)	<0.01	0.57 *
HipRTFlexion(deg)	45.15 ± 12.23(42.66 ÷ 47.64)	31.35 ± 12.32(28.73 ÷ 33.98)	<0.01	1.12 **
HipRTAbduction(deg)	19.45 ± 18.62(15.66 ÷ 23.25)	24.14 ± 3.16(23.47 ÷ 24.82)	0.29	0.43
HipRTRotationExt(deg)	−10.82 ± 18.47(−14.59 ÷ −7.06)	10.50 ± 8.11(8.77 ÷ 12.22)	<0.01	1.6 **
KneeLTFlexion(deg)	45.96 ± 12.69(43.37 ÷ 48.54)	30.84 ± 15.36(27.57 ÷ 34.12)	<0.01	1.08 **
KneeRTFlexion(deg)	60.29 ± 14.29(57.38 ÷ 63.21)	37.90 ± 12.37(35.26 ÷ 40.53)	<0.01	1.68 **
ShoulderRTRotationExt(deg)	−29.09 ± 21.80(−33.53 ÷ −24.65)	−16.20 ± 16.76(−19.77 ÷ −12.63)	<0.01	0.67 *
ShoulderRTFlexion(deg)	72.67 ± 20.92(68.41 ÷ 76.94)	71.39 ± 31.93(64.58− ÷ 8.20)	0.77	0.05
ShoulderRTAbduction(deg)	64.05 ± 35.17(56.88 ÷ 71.21)	62.28 ± 31.94(55.48 ÷ 69.09)	0.75	0.05
ElbowRTFlexion(deg)	47.69 ± 26.08(42.38 ÷ 53.00)	41.83 ± 21.71(37.20 ÷ 46.46)	0.42	0.25
WristRTExtension(deg)	8.77 ± 23.69(3.94 ÷ 13.59)	−30.71 ± 29.68(−37.04 ÷ −24.39)	<0.01	1.48 **
WristRTRadial(deg)	19.27 ± 17.28(15.74 ÷ 22.79)	−5.22 ± 17.96(−9.04 ÷ −1.39)	<0.01	1.39 **
WristRTSupination(deg)	76.47 ± 40.56(68.21 ÷ 84.74)	90.83 ± 31.36(84.15 ÷ 97.51)	0.05	0.4
ACCMax(m/s^2^)	194.79 ± 19.30(190.77 ÷ 198.81)	173.05 ± 23.82(167.82 ÷ 178.29)	<0.01	1.01 **

Data are presented as mean ± SD (Confidence intervals). Abbreviations: ACCMax—maximal acceleration of the playing hand, Ext—external, LT—left, RT—right, *p*—value of p U Mann-Whitney test, p and Cohen’s d—statistical significances: *—medium effect size, **—large effect size.

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
