# Peer review of "Gender Differences in Kinematic Parameters of Topspin Forehand and Backhand in Table Tennis"

_ijerph, 2020, doi:10.3390/ijerph17165742_

Round 1

Reviewer 1 Report

line 59-67: It is noteworthy that in martial arts, there are also differences in kinematics between women and men - WÄ…sik J, Ortenburger D, Góra T. The masurement of a taekwondo front kick Balt J Health Phys Act. 2019;11(1):76-82

line 87-88: Provide more details, e.g. Name of the Ethics Committee, consent number

line 151: Table 1 and 2. In the range can better apply the mark "÷"

Reviewer 2 Report

General Comments:

English is not written well and sufficient. The analysis is not statistically valid or does not follow the norms of the field of biomechanics studies. In addition, these values are not scientifically convincing. Also, in discussion and conclusions, the arguments are illogical and unstructured or invalid. In addition, the authors discuss differences between males and females players which wouldn't scientifically add any new information for training. 

Introduction Comments:

Insufficient information about the previous study findings is presented for readers to follow the present study rationale and procedures.

Materials and Methods Comments:

There is no clarification of a few details and provision of a rationale for the use of this particular method of measuring; these notes should be provided. In addition, there are no kinematic variables of the racket included in the experiment.

Discussion Comments:

In general, the discussion evidence is insufficient and the authors presented only the previous results without any explanation regarding the current findings.

Reviewer 3 Report

This work analyses the difference between men and women of biomechanical characteristics in two specific table tennis moves. The sample is small but enough as they are high-level performers.

The authors have carried out an interesting introduction explaining some differences between the gender of mechanical characteristics in different sports. The methods section is well explained as well as results. The discussion is poor as authors presented so much data in the results section but these data have not been deeply explained. I suggest to the authors to deepen in the explanation of these biomechanical differences between gender

51. It would be interesting if the authors added some information regarding the specific differences seen in previous studies.

53-54. Please, specify the studies that address this information.

54. These differences in fat mass are typical not only in table tennis players but also in the general population.

55. Reference 8 is incorrectly placed.

75-76. Topspin forehand and backhand should be described in the introduction.

79. Why in angular parameters? These parameters should have appeared earlier in the introduction, explaining how the difference in body composition between men and women can affect these given angular parameters.

85. men are in brackets "()" and women are not. Please, unify.148. Do you mean d≥80?

154. the symbols * and ** are not necessary because they have been explained in the statistics section in the methods.

158. Same

220. It should be deeply explained what kind of anatomical structures are responsible to provoke these differences.

232-238 Needs more explanation

Round 2

Reviewer 2 Report

Dear authors 
Thank you so much for your efforts regarding the corrections of all comments. 
Best wishes

Author Response

Responses to the Editor's and Reviewers' Comments.

Dear Editors & Reviewers of the IJERPH Journal,

we appreciate very much for all the constructive comments and useful observation. We also thank You for the effort and time put into the review of our manuscript.

Rev. 1

Dear Reviewer

Thank you very much for your comments. We checked the manuscript onse agailn for spelling. We also sent our manuscript to editing and proofreading service after the first review.

Once again, we appreciate all your comments very much.

Ziemowit Bańkosz, Sławomir Winiarski, Ivan Malagoli Lanzoni

Reviewer 3 Report

The authors have adressed all my concerns. Therefore, the article can be published in its current form.

Line 54. The reference "Bankosz and Winiarski..." is not correctly cited. The specific reference should be placed in bracets.

Author Response

Responses to the Editor's and Reviewers' Comments.

Dear Editors & Reviewers of the IJERPH Journal,

we appreciate very much for all the constructive comments and useful observation. We also thank You for the effort and time put into the review of our manuscript.

Rev. 2

Dear Reviewer

Thank you very much for your comments. We added correct number of reference in brackets

Once again, we appreciate all your comments very much.

Ziemowit Bańkosz, Sławomir Winiarski, Ivan Malagoli Lanzoni